# Defining a core configuration for human centromeres during mitosis

Ayantika Sen Gupta[1], Chris Seidel[1], Dai Tsuchiya [1], Sean McKinney[1], Zulin Yu[1], Sarah E. Smith [1], Jay R. Unruh [1] & Jennifer L. Gerton [1,2] ✉

The centromere components cohesin, CENP-A, and centromeric DNA are essential for biorientation of sister chromatids on the mitotic spindle and accurate sister chromatid segregation. Insight into the 3D organization of centromere components would help resolve how centromeres function on the mitotic spindle. We use ChIP-seq and super-resolution microscopy with single particle averaging to examine the geometry of essential centromeric components on human chromosomes. Both modalities suggest cohesin is enriched at pericentromeric DNA. CENP-A localizes to a subset of the α-satellite DNA, with clusters separated by ~562 nm and a perpendicular intervening ~190 nM wide axis of cohesin in metaphase chromosomes. Differently sized α-satellite arrays achieve a similar core structure. Here we present a working model for a common core configuration of essential centromeric components that includes CENP-A nucleosomes, α-satellite DNA and pericentromeric cohesion. This configuration helps reconcile how centromeres function and serves as a foundation to add components of the chromosome segregation machinery.

Chromosomes undergo massive structural transitions from interphase to metaphase in preparation for chromosome segregation. These transitions include resolving catenations between sister chromatids, loss of cohesion between sister chromatids and nearly 10,000-fold compaction of the DNA[1,2]. These structural transitions are coordinated by SMC complex proteins – cohesins and condensins. While most cohesin is lost from the arms of sister chromatids by metaphase, it is retained at the centromeres until early anaphase[3]. Cohesion at centromeres is essential to counterbalance forces from kinetochore microtubules and ensure bi-orientation of all sister chromatids prior to segregation[4–6]. The cohesin protein complex is responsible for sister chromatid cohesion. It is composed of 3 subunits—SMC1, SMC3 and RAD21, that form a ring with a predicted diameter of 50 nm[7]. The fourth subunit of this complex is the accessory stromal antigen (SA) subunit. Human somatic cells express SA1 and SA2 subunits that form mutually exclusive cohesin complexes. The SA2-cohesin complex is suggested to maintain centromeric cohesion while the SA1-cohesin complex is suggested to be responsible for telomeric cohesion during mitosis, based on functional outcomes of knockdown[8]. While the higher order organization of chromosome arms by SMC complex proteins has been investigated[9–11], the organization of human centromeres remains elusive.

Human chromosomes have regional centromeres composed of highly repetitive α-satellite DNA that forms higher order repeat (HOR) arrays, a subsection of which harbors the histone H3 variant CENP-A[12]. CENP-A enriched chromatin creates a foundation for building the kinetochore. Plus ends of kinetochore microtubules capture chromosomes by their interactions with outer kinetochore proteins at the centromere[13]. Due to the dynamic instability of the plus ends of microtubules, centromeres of sister chromatids are pulled apart when microtubules attach and depolymerize, and then recoil upon microtubule detachment or repolymerization. This centromere stretching can occur over micron sized distances and multiple times during metaphase[4]. The 50 nm cohesin ring cannot directly encircle CENP-A containing centromeric DNA because it is inconsistent with the observed stretching behavior in mitosis. While theoretical models for this behavior have been proposed, especially for budding yeast point centromeres[14–17], the field currently lacks data-informed models for

[1]Stowers Institute for Medical Research, Kansas City, MO, USA. [2]Department of Biochemistry and Molecular Biology, University of Kansas, Kansas City, KS, USA. ✉e-mail: jeg@stowers.org

regional centromeres that can accomodate the two essential processes of kinetochore-microtubule attachment, and sister chromatid cohesion.

Human centromeres are highly polymorphic, both in terms of their size and sequence. Part of this polymorphism is due to centromere copy number variations because of their repetitive nature. However, inter-kinetochore distances in mitosis are surprisingly conserved not just within humans but range from 800 to 1000 nm across organisms[14]. This suggests that although the underlying linear size and sequence composition of centromeric DNA is highly variable, centromeres share similar physical features. It is unclear how cohesin accommodates variations in centromeric DNA sequence and contributes to the formation of a core centromere structure on mitotic chromosomes.

Recent advances in super-resolution microscopy enabling visualization of SMC complex proteins[18–21] and the linear assembly of human centromeric DNA[12], create an exciting opportunity to significantly advance our understanding of centromere geometry. In this study, we present the binding landscape of cohesin at human centromeres, based on newly available centromere assemblies. We combine this information with super-resolution microscopy to determine the localization of cohesin relative to CENP-A bound chromatin within centromeric α-satellite HOR arrays. Integrating the data provides insights regarding organization of human centromeres in 3D during mitosis and how centromeric DNA size variation is accommodated to build a core centromere geometry.

## Results

### Cohesin binding landscape at human centromeres

To begin to reconcile centromeric cohesion with the observed separation of human centromeres up to 1–2 μm during mitosis, we examined the binding landscape of cohesin at centromeres of all human chromosomes. We performed chromatin immunoprecipitation (ChIP) for cohesin subunits RAD21, SA1 and SA2 in hTERT CHM13 (complete hydatidiform mole) and hTERT RPE-1 (retinoid pigmented epithelium) cell lines. Paired-end sequencing (depth−10–40 million reads) was performed and reads were mapped using an established algorithm[22]. To evaluate the ChIP-seq data, we aligned reads from CHM13 and RPE-1 datasets to the T2T-CHM13-v1.0 reference[12], for which complete linear maps of centromere sequences are assembled and available (Fig. 1a). Next, we determined genome-wide binding sites for RAD21, SA1 and SA2 proteins in CHM13 and RPE-1 cells. We compared the overlap of RAD21 peaks in CHM13 cells with CTCF binding sites obtained from publicly available datasets[23] and found that over 70% of CTCF sites overlapped with RAD21 binding sites (Fig. 1a), suggesting that cohesin was similarly enriched at CTCF binding sites and TAD boundaries, as previously reported[24].

We next examined enrichment of cohesin within centromeres. Centromeres on human chromosomes are composed of large arrays of higher order repeats (HOR) of α-satellite DNA[25–27]. Several human chromosomes contain multiple α-satellite repeat arrays. However, only a single array in each chromosome is bookmarked by the histone H3 variant CENP-A[28]. This is termed the "active" α-satellite array, where the kinetochore complex assembles. Centromeric regions are refractory to meiotic recombination leading to their inheritance across generations as centromere haplotypes (CenHaps)[29–31]. CenHaps have been defined for humans and encompass the active α-satellite array, inactive arrays, and extend into surrounding pericentromeric DNA. We used previously described mapping parameters and CENP-A ChIP data from CHM13[22] to recapitulate that CENP-A mapped to the "active" α-satellite arrays on all chromosomes (Fig. 1b) but was highly enriched within a subdomain termed the centromere-dip region (CDR), defined by low CpG methylation[32]. CENP-A mapped exclusively to the "active" α-satellite HOR array in each CenHap in both CHM13 cells (Fig. 1b–d) and in RPE-1 cells (Fig. S1a, S1b). We used identical parameters to map

RAD21, SA1 and SA2 to the genome and examined the enrichment profile within CenHaps. CenHaps and α-satellite HOR arrays comprise 10.1 and 2% of the total CHM13 genome. In contrast, only 5.2 and 1.25% total RAD21 reads mapped to CenHaps and active HOR arrays respectively in CHM13 cells, suggesting these regions are relatively depleted for cohesin (Fig. 1e). Cohesin peaks per megabase within α-satellite arrays were negligible in comparison to randomly selected regions of the genome (Figs. 1b–e, S1a-S1b).

We next assessed cohesin peak density within the larger CenHap domains that include active and inactive α-satellite HOR arrays and pericentromeric DNA. Overall, peak density of cohesin was lower within CenHaps in comparison to the rest of the genome. Strikingly, cohesin enrichment was limited to the pericentromeric domains of CenHaps (Figs. 1b–d, S1a-S1b). Lower enrichment of cohesin within CenHaps in comparison to the rest of the genome could partly be due to fewer CTCF binding sites and low transcription within these regions (Fig. 1f, S1c, Table S1), factors that are normally associated with cohesin binding[33–35]. We examined cohesin binding within 2 Mb regions flanking the CenHaps (CenHap Adjacent) and found cohesin peak densities similar to the rest of the genome (Fig. 1c, d, S1a-S1b). Our results reveal that α-satellite HOR arrays are highly enriched in CENP-A but are largely depleted of cohesin. Instead, cohesin associates with pericentromeric regions within CenHaps. Thus, DNA regions responsible for two major functions of centromeres – building a kinetochore and sister-chromatid cohesion, are integrated within the CenHaps but in non-overlapping domains.

### 3-dimensional organization of cohesin at centromeres during mitosis

To extend the linear binding profiles and determine how cohesin contributes to organization of centromeres in 3-dimensional (3D) space, we examined cohesin localization at centromeres of mitotic chromosomes. For this, we performed confocal and 3D-Structured Illumination Microscopy (SIM) on mitotic chromosome spreads and examined cohesin localization relative to CENP-A. Chromosome spreads were generated by arresting cells in mitosis with colcemid which depolymerizes microtubules. Hence, centromeres are represented without microtubule forces. Confocal microscopy revealed two CENP-A foci on each chromosome, denoting the active α-satellite DNA on the two sister chromatids (Fig. 2a). RAD21 signal was highly enriched in a linear orthogonal axis between CENP-A clusters (Fig. 2a), similar to the localization of RAD21 observed in *Drosophila* S2 cells[36]. A few chromosomes in every mitotic spread retained patches of cohesin on chromosome arms visible as discontinuous signals between sister chromatids. Capturing the RAD21 signal from mitotic chromosomes allowed us to evaluate its position relative to CENP-A clusters and α-satellite DNA in 3D.

To further resolve the RAD21 signal proximal to the centromere, we performed 3D-SIM which provides an 8-fold volumetric increase in resolution over conventional light microscopy. SIM improved the resolution of RAD21 and CENP-A signals and allowed us to identify finer structures of metaphase chromosomes demonstrating (1) spatial separation of RAD21 signals from CENP-A and (2) a block of RAD21 parallel to the axis between the sister chromatids, which we presume corresponds to the pericentromeric cohesin observed by ChIP (Fig. 2b). SMC1A endogenously tagged with mEGFP displayed similar localization as RAD21, confirming the pattern for a second subunit of the cohesin complex (Fig. S2a, b). We observed variation in cohesin localization across chromosomes from the same spread as demonstrated by the two examples shown (Fig. 2b). However, identification of commonalities would enable us to define the core geometry of centromere components.

To generate average representative positions of RAD21 and CENP-A, we employed a single particle averaging pipeline[37], where hundreds of chromosomes were aligned to generate an average

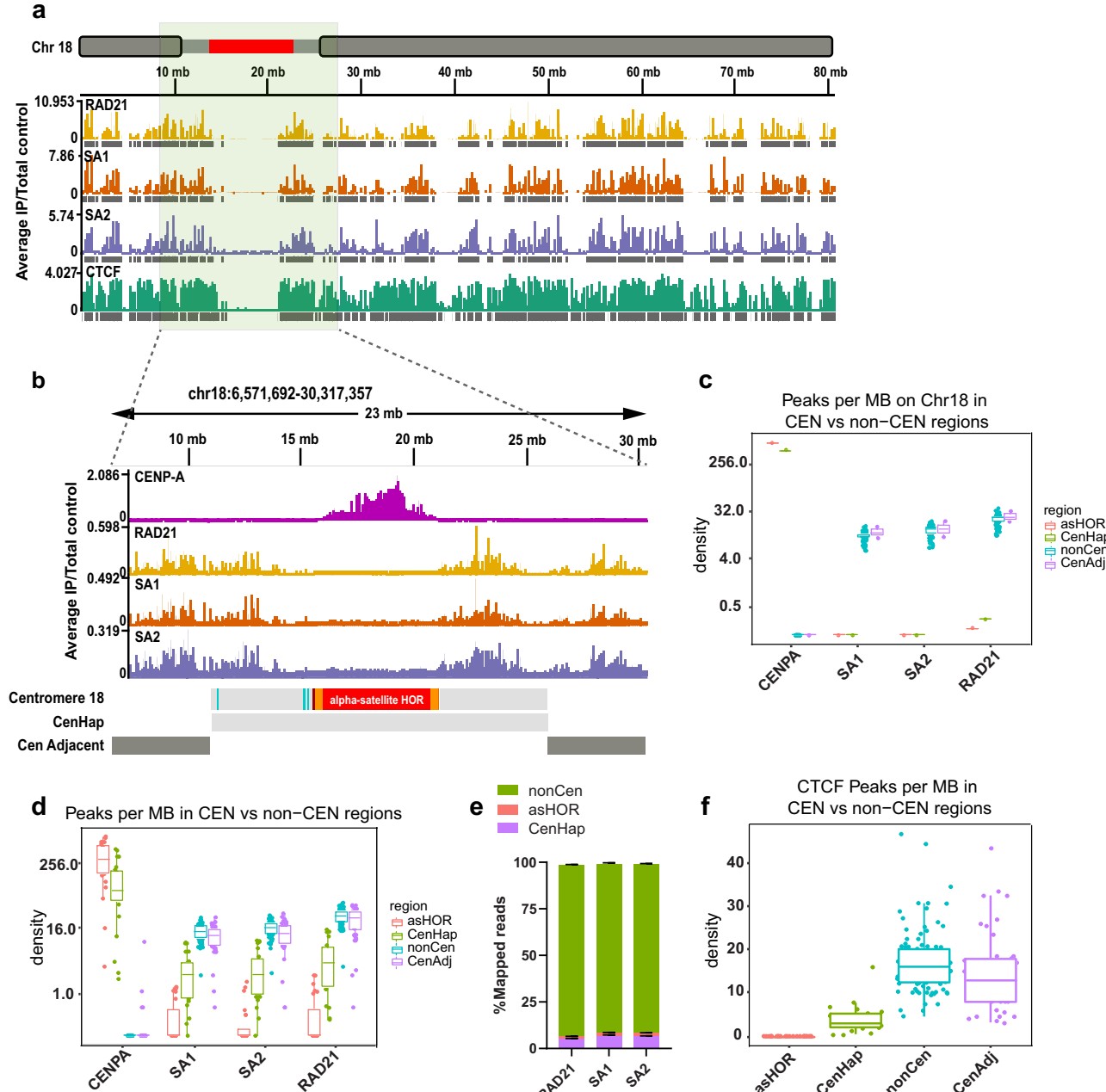

**Fig. 1 | Cohesin is enriched at human pericentromeres but not α-satellite HOR arrays. a** Linear binding profile of three cohesin subunits - RAD21, SA1, SA2 and CTCF, mapped to T2T CHM13 v1.0, across the length of chromosome 18 is displayed. Centromere haplotype (CenHap) of chromosome 18 denoted in red. Peaks called by macs are denoted as gray tracks. Data is an average representation of 3 biological replicates. **b** Zoomed in view of centromere 18 CenHap with 10 Mb flanking regions (Cen Adjacent) is shown. CENP-A ChIP peaks are enriched within the D18Z1 α-satellite HOR array (red). Cohesin subunits lack binding within the α-satellite HOR array but are enriched outside of the HOR array in the pericentromeric domains within the CenHap. **c** Peak densities (number of peaks per Mb) of CENP-A, SA1, SA2 and RAD21 within active α-satellite HOR array (asHOR), CenHap, 2 Mb regions flanking CenHap (CenAdj) and 100 random genomic regions outside

of the centromere haplotype (nonCen) of chromosome 18 in CHM13 cells is shown. **d** Metagenome analysis of peak densities (log₂) of CENP-A, SA1, SA2 and RAD21 within the above stated regions, across all chromosomes is shown. **e** Mean proportion of mapped reads for RAD21, SA1 and SA2 in nonCen, asHOR and CenHap regions of the CHM13 genome is shown. Data show mean of 3 biologically independent replicates. Error bars denote S.D. **f** Peak densities (log₂) of CTCF in the above stated regions of the CHM13 genome mirroring peak densities of cohesin factors. For all box plots, upper and lower whiskers represent the largest and smallest values no further than 1.5 * inter quartile range (IQR) of the bounds, respectively. Data outside of whiskers are outlying points. Center denotes median, and lower and upper bounds of box denote 25th and 75th percentile of peak densities, respectively.

image of RAD21 and CENP-A (Fig. 3a). We next defined a lateral axis by joining the centers of the CENP-A clusters on sister chromatids and assessed CENP-A and RAD21 intensity along this axis from the summed image (Fig. 3b). From the mean intensity profile, we find that the lateral CENP-A peak to peak distance is 562 nm (±91.6 nm S.D.) on native centromeres without tension (Fig. 3c). The inter-

CENP-A distance recorded by us, denoting distance between centromeric chromatin on sisters, is narrower than inter-CENP-T distance of ~800 nm reported previously, denoting inter-kinetochore distances[38], consistent with what is known about kinetochore structure. The size of the CENP-A cluster on each sister chromatid was 81 nm (S.D.). RAD21 showed three intensity peaks along the lateral

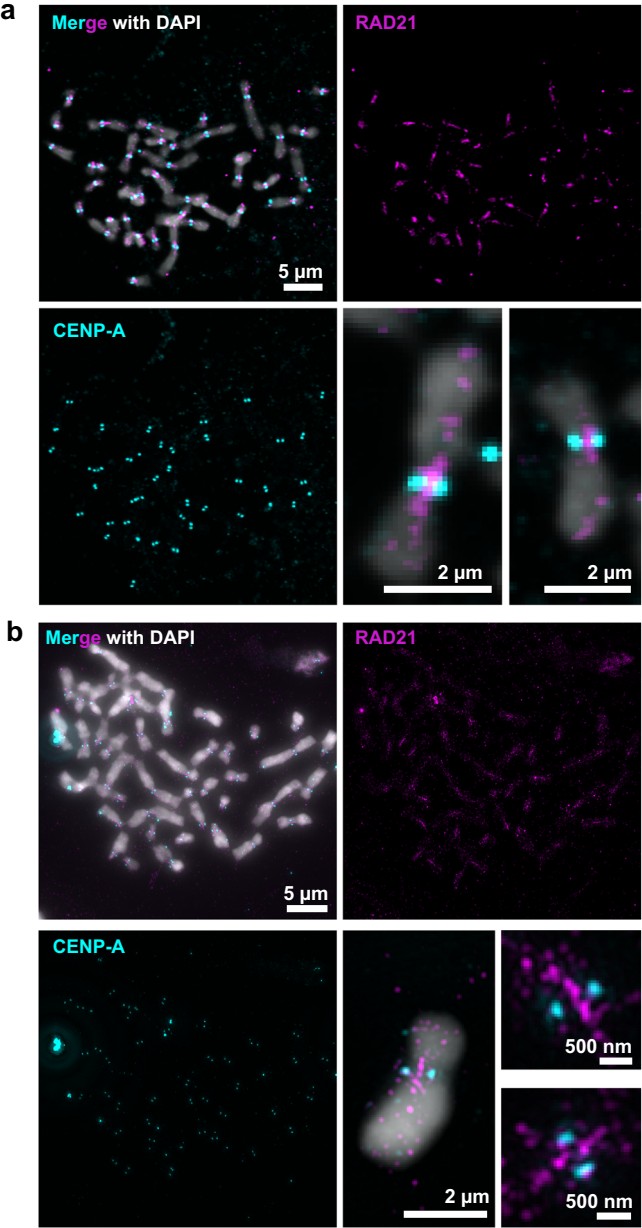

**Fig. 2 | Cohesin and CENP-A domains are non-overlapping on centromeres of mitotic chromosomes. a** Confocal microscopy images of native mitotic chromosomes immunostained for CENP-A (cyan) and cohesin subunit RAD21 (magenta) are shown. Zoomed images of two individual chromosomes show enrichment of CENP-A and RAD21 at the centromere and retention of some RAD21 on chromosome arms between sister chromatids. **b** 3D-Structured illumination microscopy (SIM) images show RAD21 is enriched in an axis orthogonal to CENP-A and localizes between sister chromatids. Zoomed images show non-overlapping CENP-A and RAD21 signals. Data are representative of more than 10 chromosome spreads (*n* > 400 total chromosomes).

axis, with the strongest enrichment at the center of the lateral axis (Fig. 3d). The lateral width (S.D.) of the RAD21 signal between the two sister chromatids was 190 nm (Fig. 3d) and that recorded for SMC1A-mEGFP was 154 nm (Fig. S2c-e). The lateral width of the cohesin signal, as observed by us, was narrower than the centroid to centroid distance (~750 nm) of mitotic sister chromatids, as demonstrated in a previous study[11]. The pattern we identify is consistent with the central axial cohesin being localized between sister chromatids. Averaged images also showed two minor peaks of RAD21 ~100 nm outside the center of CENP-A clusters. Similar signals for SMC1A have been

detected using signal amplification methods in the human colon cancer cell line HCT-116[21]. However, we did not detect CENP-A proximal SMC1A-mEGFP enrichment using immunostaining (Fig. S2b, SMC1A-mEGFP lateral fit). Given the relative depletion of cohesin in α-satellite DNA observed by ChIP (Fig. 1e), these data suggest that a small labile fraction of cohesin exists at these sites that can only be detected by signal amplification. 3D-SIM allowed us to appreciate that RAD21 and CENP-A are enriched at non-overlapping domains on mitotic chromosomes.

We further explored the RAD21 signal between sister chromatids along the chromosomal axis. Averaged intensities showed an axis extending ~800 nm but with a significant dip in the middle at the intersection of the lateral and chromosomal axes (Fig. 3e). This dip indicates a cohesin depleted zone at the center of the structure (Fig. 3e) and would not be detectable without super-resolution imaging. Taken together, linear and 3D localization of cohesin and CENP-A reveal mutually exclusive CENP-A and cohesin enriched domains within CenHaps. The distance between CENP-A clusters combined with mapping of CENP-A to active α-satellite DNA strongly suggests active α-satellite DNA does not participate in centromeric cohesin-based cohesion. The localization of cohesin between sister chromatids along the chromosomal axis is more consistent with its enrichment on pericentromeric domains, also seen with ChIP-seq analysis (Fig. 1c).

**Classes of cohesin organization at centromeres**

As noted in Fig. 2, there is variability in cohesin signals across different chromosomes within a spread. We further explored this variation specifically at centromeres, where some chromosomes displayed a cohesin-free region at the center. To evaluate these differences, we quantified the ratio of RAD21 intensity at the center versus at a point 240 nm (6 pixels) away from the center along the chromosomal axis (Fig. 4a) and sorted centromeres in ascending order of this ratio (Fig. 4b). While this analysis showed a continuum of spatial distributions of cohesin along the chromosomal axis, the organization of cohesin at one end of this continuum (center depleted) versus the other (center enriched) was obvious (Fig. 4b). We arbitrarily defined three classes of organization—(1) strong enrichment of RAD21 at the intersection of the lateral and chromosomal axes (central), (2) even enrichment of RAD21 along the chromosomal axis extending towards the chromosome arms from the center (uniform), and (3) lower enrichment at the center but stronger intensity extending towards the arms (split) (Fig. 4b). We also obtained the single particle average for each class and constructed intensity profiles along the chromosomal axis that clearly demonstrated the three individual classes of cohesin organization at the centromeres (Fig. 4c, d). Each metaphase spread contained all three classes of RAD21 enrichment (Fig. S2f) indicating that the classes were unlikely due to differential chromosome condensation across spreads. We asked if inter-CENP-A distances were distinct for each class. The inter CENP-A distances between the three RAD21 classes—split (570 nm), uniform (554 nm) and central (557 nm), were not significantly different, suggesting that the variability in RAD21 localization patterns does not dictate this parameter (Fig. 4e).

We speculate that the three classes correspond to chromosome-specific variation in cohesin binding and organization within pericentromeric DNA. The classes are helpful but artificial since there is a continuum of variation in the RAD21 signal. This is consistent with the fact that each chromosome bears a unique CenHap. While the identities of each chromosome and the origins of the different cohesin enrichment patterns are unknown, the subtle differences in inter-CENP-A distance may indicate chromosome-specific centromere organization. Overall, these data allow us to glean the average arrangement of cohesin and CENP-A at centromeres while also suggesting the possibility for centromere-specific organization.

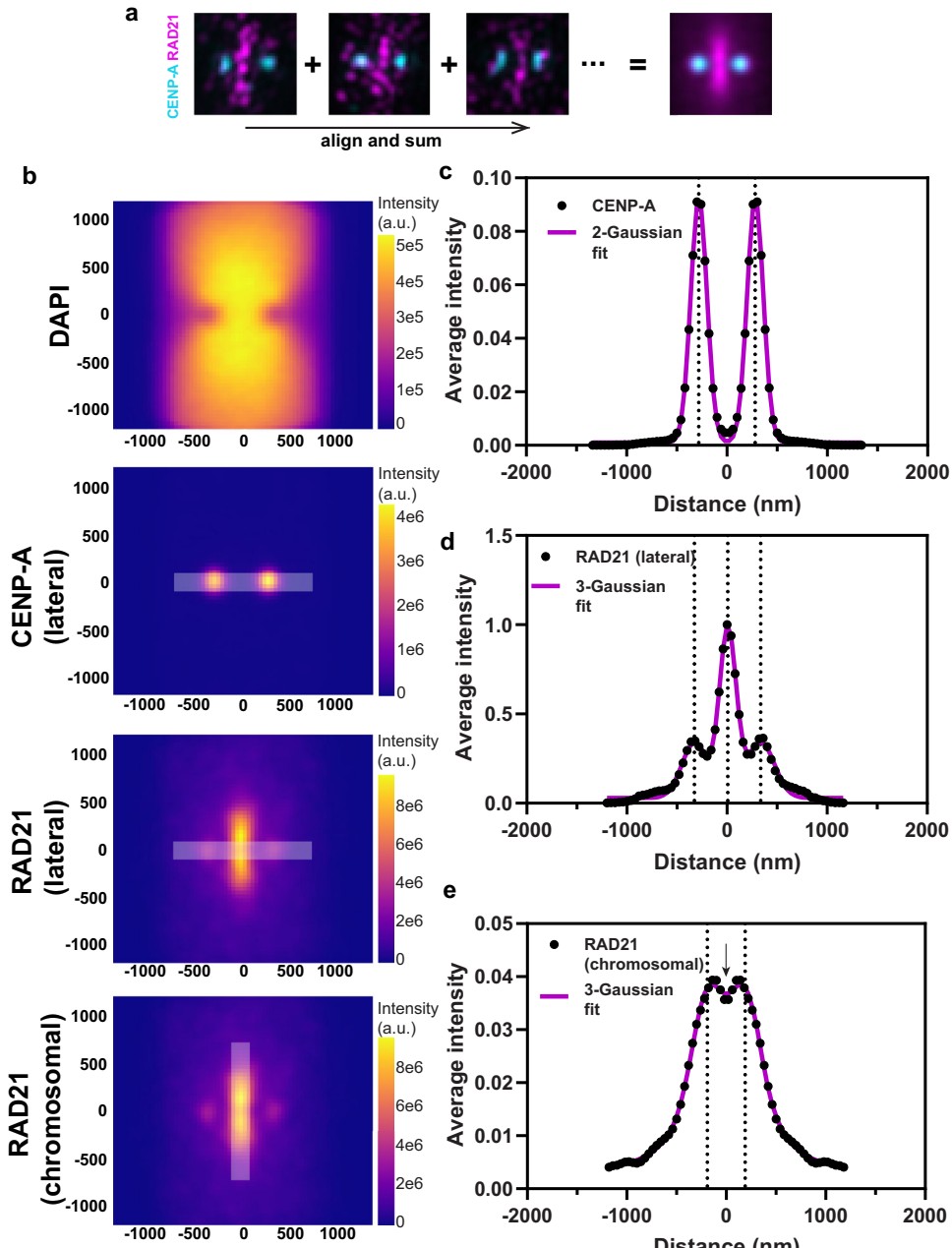

**Fig. 3 | Single particle averaging of 3D-SIM images with aligned CENP-A and RAD21 signals. a** CENP-A paired clusters were identified with an automated algorithm and their centers were used to align all centromeres. Centroids of CENP-A signals were joined, and the center of this line was defined as the center of each centromere. 60px × 60px surrounding 463 individual centromeres (18 spreads) were cropped and signals from all images were summed. **b** Summed intensities of DAPI, CENP-A and RAD21 around centromeres are shown ($n = 463$ chromosomes). x and y axes indicate distance (nm). **c–e** Intensities (white overlay line in (**b**)) of CENP-A measured along the lateral axis and RAD21 measured along the lateral and along the chromosomal axes are shown (black closed circles in (**c–e**)), along with their Gaussian fits (magenta line). Dotted lines represent positions of peaks from the fits. Arrow indicates the dip in RAD21 intensity at the center of the vertical axis.

## Visualization of centromeric α-satellite DNA during mitosis

To expand our understanding of the organization of centromeric α-satellite DNA on mitotic chromosomes, we attempted to co-visualize RAD21 and α-satellite DNA using immuno-FISH. However, the anti-RAD21 antibody was incompatible with the DNA denaturation conditions required for FISH. Alternatively, since our single particle averaging could generate average positions of CENP-A clusters with a precision of 83 nm, we used CENP-A as a fiducial landmark to position cohesin with respect to other DNA markers. To visualize centromeric DNA we used two approaches—firstly we used a FISH probe to the CENP-B box as a general marker for α-satellite DNA on all chromosomes. CENP-B boxes are 17 bp motifs that are present throughout all

α-satellite DNA except on the Y-chromosome[39]. Immuno-FISH labeling of CENP-A and CENP-B boxes on native mitotic chromosomes showed that α-satellite DNA on all chromosomes spans the lateral zone between CENP-A clusters (Fig. 5a), with multiple different symmetric and asymmetric configurations (Fig. 5a, zoom).

To assess the average position of α-satellite DNA relative to CENP-A, we performed single particle averaging of SIM images (Fig. 5b). Measurements across the lateral axis showed that CENP-A clusters partially overlapped with the α-satellite signal (Fig. 5c), positioned at the outer edge of the structure, where the kinetochore forms and attaches to microtubules. This is consistent with CENP-A nucleosomes densely occupying a subregion of each α-satellite domain[40,41]. CENP-B

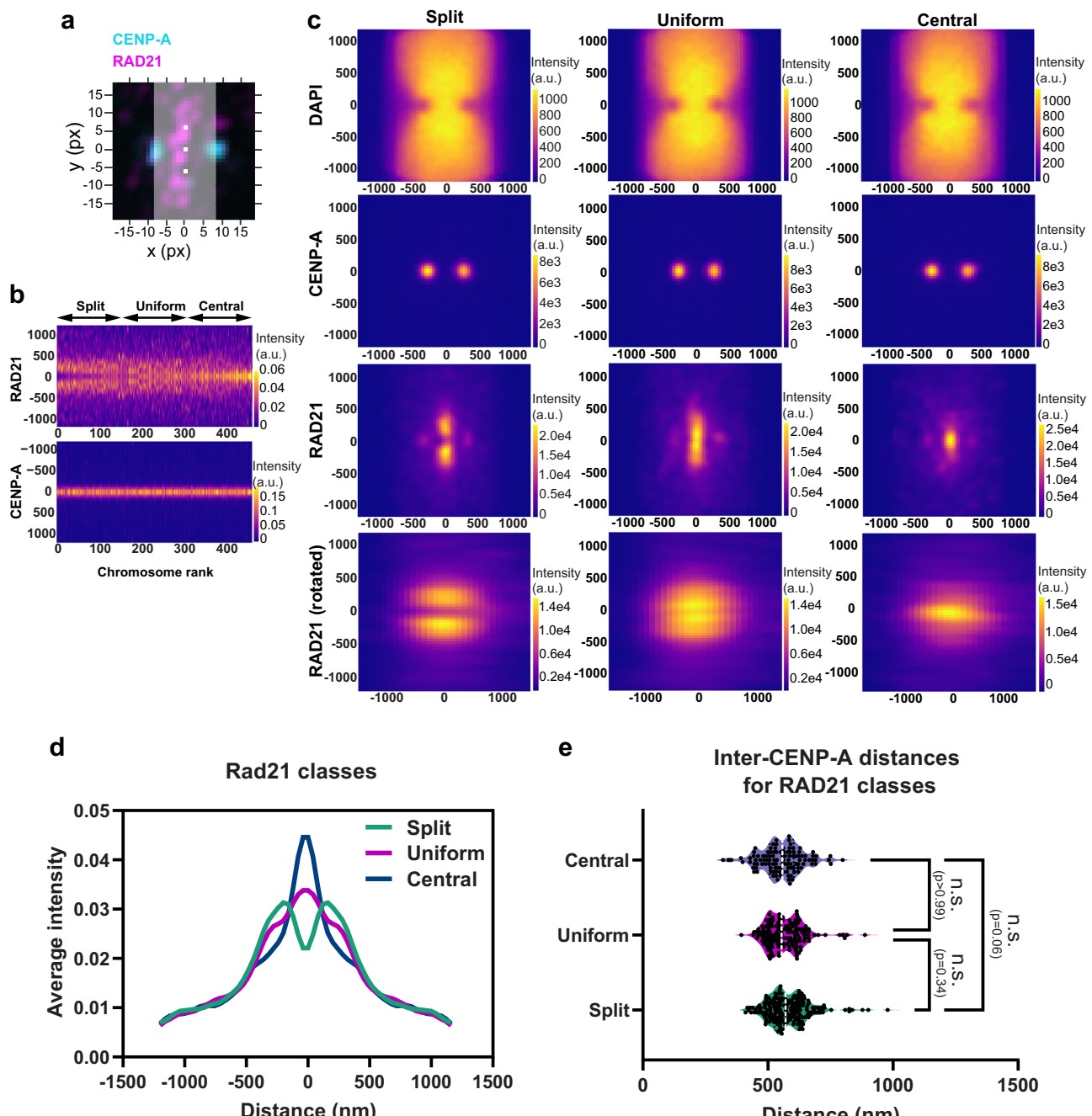

**Fig. 4 | Sub-classification of centromeres by cohesin localization. a** A representative centromere with CENP-A (cyan) and RAD21 (magenta) signals and the line profile (gray) used for measurements is shown. Center-enriched RAD21 ratio was determined by measuring the ratio of intensity at pixels (white) at the center of the centromere and 6 pixels vertically away from the center. **b** Chromosomes were assigned ranks based on center-enriched RAD21 ratio (Ratio 0.0861–Rank 1, Ratio 5.519–Rank 463). A montage is shown in which the x axis represents each centromere, and the y axis represents the vertical profile through the middle of the centromere region, sorted by their rank. Three classes–split (ranks: 1–200), uniform (ranks:201–350) and central (ranks:351–463) were arbitrarily assigned based on these ratios. **c** Aligned and averaged images for each class are shown. x and y axis on images indicate distance (nm). **d** Average intensities of RAD21 along the vertical profile in the three classes are shown. **e** Inter-CENP-A distances measured for the three classes–central ($n = 200$), uniform ($n = 150$), split ($n = 113$) are compared by unpaired one-way Kruskal-Wallis test (multiple comparisons). Dotted white lines in violin plots denote median values. n.s. - not significant.

FISH signals were broadly distributed with weak peaks at 209 nm from the center of the structure and a shallow dip (411 nm wide) between the two sister chromatids (Fig. 5c). Peak intensities of CENP-B box containing DNA lie inside of CENP-A peaks at 300 nm on either side (Fig. 5c). The thickness of the α-satellite DNA suggested that it is organized in the form of one or more loops with CENP-A nucleosomes positioned at the microtubule-proximal tips. The pericentromeric

cohesin intensity (190 nm wide, Fig. 3b) is positioned exactly within this dip region of the α-satellite DNA and potentially extrudes a loop of α-satellite DNA, which likely has additional folding and organization based on its appearance.

While CENP-B box FISH highlighted the average locations of α-satellite DNA across all chromosomes, we wanted to investigate positional variation in the context of a single centromere. For this we chose

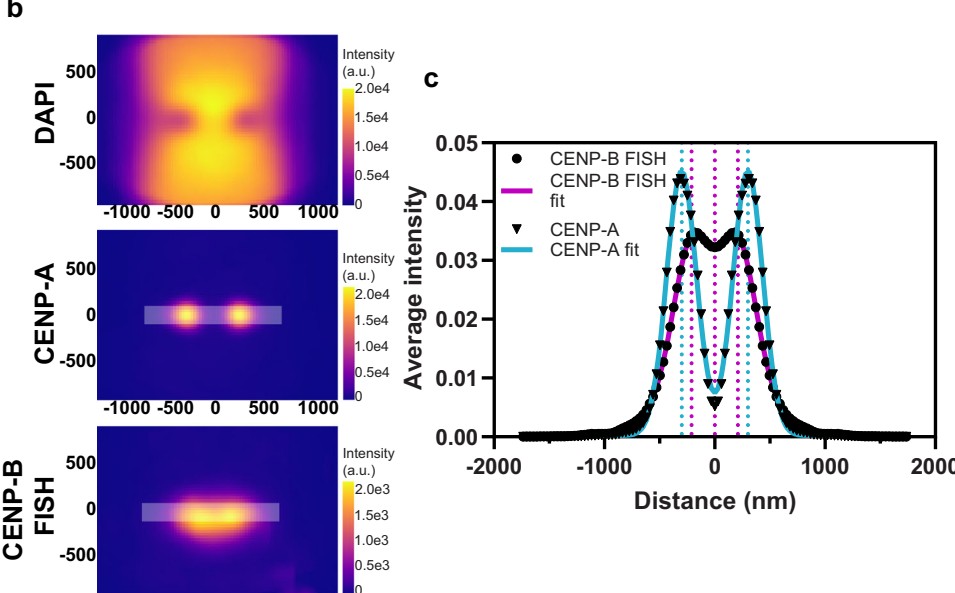

**Fig. 5 | Single particle averaging of 3D-SIM images with aligned CENP-A and CENP-B FISH signals. a** SIM images of native mitotic chromosomes display localization of centromere components CENP-A (cyan) by immunostaining and α-satellite DNA by CENP-B box FISH (magenta). Zoomed images of two individual centromeres show α-satellite DNA visualized by CENP-B box FISH bridging CENP-A clusters. Experiment was performed once but reproducible across n = 13 metaphase spreads representing 494 individual chromosomes. **b** Summed intensities of DAPI, CENP-A and CENP-B FISH around centromeres from n = 494 chromosomes are shown. x and y axes indicate distance (nm). CENP-B FISH signals are slightly bowed. **c** Average intensities of CENP-A and CENP-B FISH signals are shown along the lateral axis as indicated in (**b**) along with their Gaussian fits (bold lines). Dotted lines represent positions of peaks from the fits.

the active α-satellite HOR array on chromosome 7−D7Z1 (Fig. 6a). Very similar to general α-satellite DNA enrichment patterns across all chromosomes, as seen by CENP-B FISH (Fig. 5a), the D7Z1 α-satellite HOR array occupied the area between the CENP-A foci (Fig. 6b). SIM of D7Z1 revealed a structure with an extensive spread along the chromosomal (247 nm) and lateral (peaks at 206 nm on either side of the center, S.D. −175 nm) axes, and a dip in signal in the center (Fig. 6c). As previously observed for averaged signals across all centromeres, CENP-A intensity partially overlapped with D7Z1 signals (Fig. 6c). Average inter-CENP-A distances for centromere 7 was 634 nm (±102 nm) (Fig. 6c), which was wider than the average inter-CENP-A distance (562 ± 91.6 nm) across all chromosomes taken together (Fig. 3c). This could point to chromosome-specific variation in inter-CENP-A distances as also seen in Fig. 4e.

The two D7Z1 signals on homologs of chromosome 7 were non-equivalent in every mitotic spread assessed, with one homolog showing higher intensity (bright) than the other (dim). G2-arrested RPE-1 cells assessed by confocal microscopy demonstrated that the dim signal of D7Z1 was nearly 50% lower in intensity than the bright signal (Fig. S3), representing distinguishable haplotypes. The differences in intensities of the haplotypes made it possible to sort D7Z1 signals in ascending order of their signal intensities (Fig. 6d). We were curious to examine the structural arrangements of two versions of the same α-satellite HOR array (D7Z1) varying in copy number. To this end, we performed single particle averaging of the dim and bright classes of D7Z1 (Fig. 6e). This showed a larger spread of α-satellite DNA along the chromosomal axis for the bright class (320 nm) in comparison to the dim class (198 nm). However, the

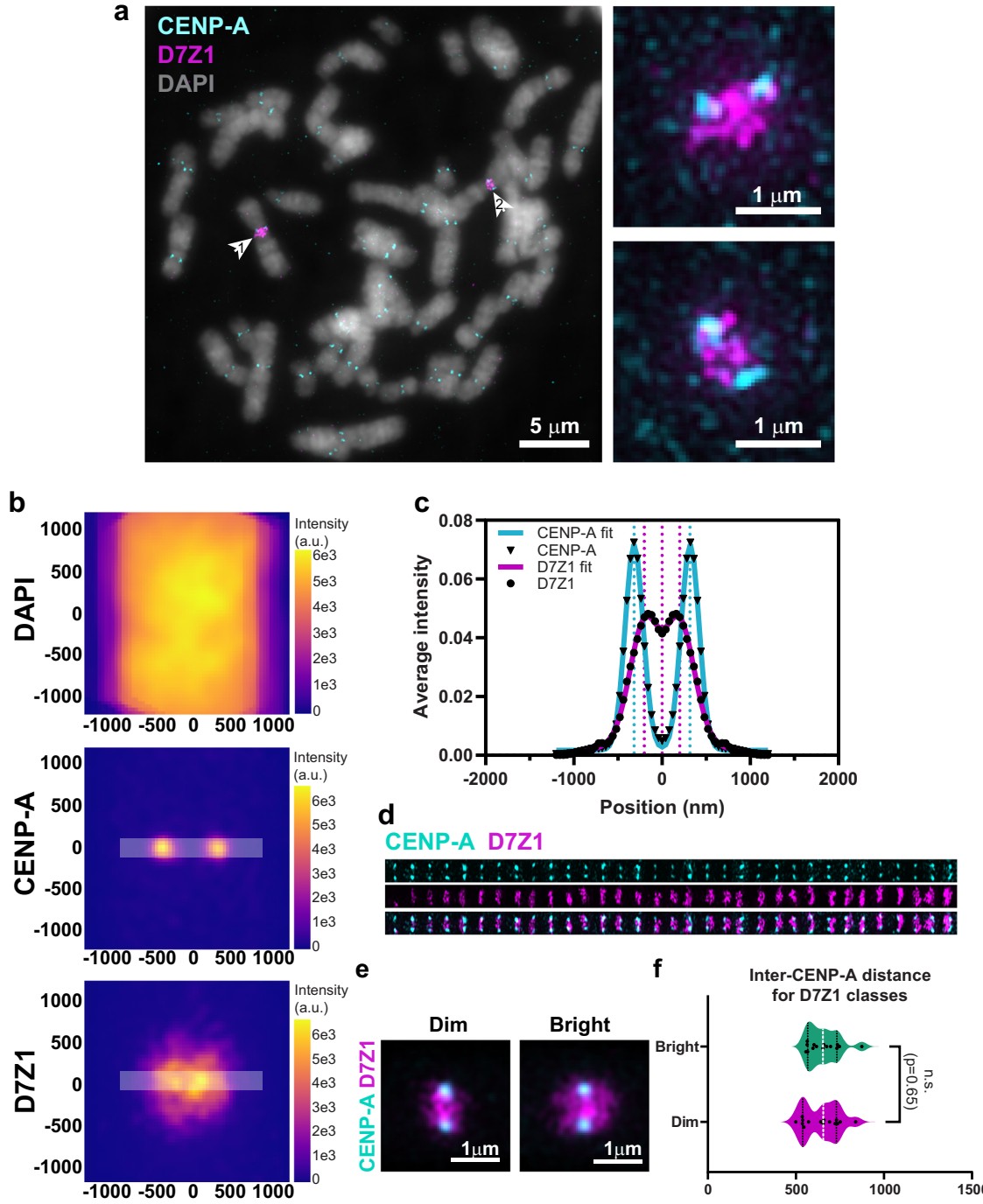

**Fig. 6 | D7Z1 copy number variants have indistinguishable inter-CENP-A distances. a** SIM images of native mitotic chromosomes for co-visualization of centromeres by CENP-A (cyan) immunostaining and centromere 7 α-satellite (D7Z1) DNA (magenta) by FISH are shown. The mitotic spread from a single RPE-1 cell shows two chromosome 7 homologs with distinct D7Z1 intensities, one bright (arrow 1) and one dim (arrow 2). Zoomed images of the bright (upper) and dim (lower) D7Z1 arrays. Experiment was performed twice and reproducible across *n* = 16 metaphase spreads representing 33 individual chromosomes (one spread was aneuploid for chromosome 7). **b** Summed intensities of DAPI, CENP-A and D7Z1

around centromeres from *n* = 33 chromosome 7. x and y axes indicate distance (nm). **c** Average intensities of CENP-A and D7Z1 FISH signals (black closed circles) measured along the lateral axis as shown in (**b**), are indicated along with their Gaussian fits (bold lines). Dotted lines represent positions of peaks from the fits. **d** Montage of 33 individual centromeres from 16 spreads, sorted by their D7Z1 intensities. **e** Aligned and averaged images are shown for the dim and bright D7Z1 classes. **f** Inter-CENP-A distances measured for the two D7Z1 arrays are compared by unpaired one-way Mann-Whitney test. n.s.—not significant.

inter-CENP-A distances from the bright and dim classes of D7Z1 were not significantly different (Fig. 6f). This suggests that while the linear sizes of the bright and dim classes of D7Z1 are significantly different, the overall core geometry of CENP-A position is conserved for the two haplotypes. This also suggests that the organization of D7Z1 α-

satellite DNA must be flexible to accommodate an overall conserved centromere geometry.

We have defined several properties of the core geometry of human centromeres. We show that overall inter-CENP-A cluster distances are similar between native human centromeres but may show

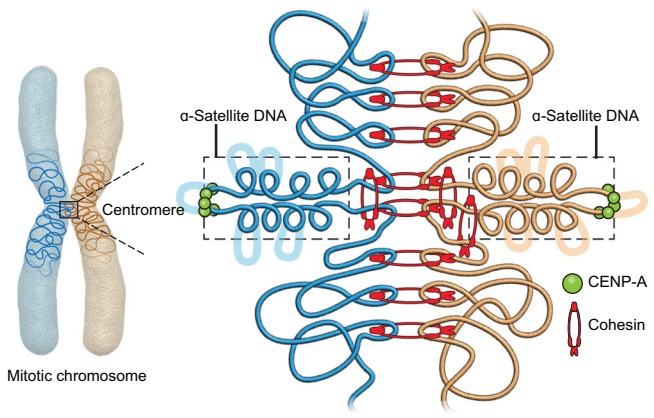

**Fig. 7 | Model for the geometry of human centromere components.** Centromeric DNA on mitotic human chromosomes is organized in 3D, with pericentromeric cohesive domains and a kinetochore-assembly domain composed of α-satellite DNA. We speculate α-satellite DNA from each sister forms a primary extensible loop with CENP-A enriched chromatin at its tip. This loop may form as a consequence of loop extrusion by cohesin at its base. The primary α-satellite loop may contain several minor loops which can be small or large (dotted lines) depending on the size of the α-satellite HOR array in a bottlebrush configuration. These may be maintained by additional structural proteins, such as condensins. The extensible loop must be able to withstand microtubule forces and thus is potentially tethered to the pericentromeric DNA by cohesin or other structural proteins. The CENP-A associated tip of the α-satellite DNA may be further organized by cohesin, observed variably by imaging and not by ChIP.

chromosome-specific variation. Cohesin forms an axis between two sister chromatids based on binding in the pericentromeric domains but is depleted from the active α-satellite DNA. Finally, changes in copy number of α-satellite DNA are accommodated, presumably by altering folding and organization while retaining a common spacing of CENP-A clusters.

## Discussion

Our study has illuminated the geometry of CENP-A at α-satellite DNA and cohesin at the pericentromeric region of human centromeres, advancing our understanding of how human centromeres operate. Centromeres must provide two key functions – microtubule attachment and centromeric cohesion. We find that both these functions are encoded in a single CenHap but are physically separated, reminiscent of similar physical separation of CENP-A centromeric nucleosomes and pericentromeric cohesin observed in budding yeast[17,42]. We find that the CenHaps harbor two distinct domains—the domain of kinetochore formation embedded in the CENP-A enriched α-satellite higher order repeat (HOR) arrays, and the domain for cohesin which is largely pericentromeric. With this arrangement the α-satellite DNA is free to stretch while cohesion is maintained at pericentromeric regions. It has been proposed that all 16 yeast centromeres together create a molecular spring on the mitotic spindle that mimics a single human centromere[14]. While our study does not examine chromosomes on the spindle, the positions of CENP-A, cohesin, and centromeric DNA suggest a model for how centromeric cohesion and stretching motions during microtubule polymerization and depolymerization in metaphase could be accommodated (Fig. 7). Our work extends beyond the point centromere structure in budding yeast and suggests a resolution for this conundrum for regional human centromeres.

Results from our ChIP-seq experiments indicate that cohesin is localized to pericentromeric domains (Fig. 1). The stromal antigen (SA) subunits of cohesin – SA1 and SA2 have been previously shown to differentially regulate telomeric and centromeric cohesion, respectively[43]. Hence, our expectation was to find preferential enrichment of SA2-cohesin within the CenHaps. Contrary to our expectation, there was no obvious predominance of SA2 at the pericentromeres. We surmise that both SA1- and SA2-cohesin complexes exist at pericentromeric regions and could both provide inter-sister cohesion. This speculation is supported by the distinction in phenotypes for loss of function for RAD21 versus SA2, wherein loss of RAD21 leads to complete loss of cohesion between sister chromatids[44], as opposed to loss of SA2-cohesin, which only leads to partial loss of cohesion at centromeres. Our observations suggest two non-exclusive possibilities: (1) SA2 complexes form more sister chromatid cohesion at pericentromeric regions, and SA1 binds but provides little cohesion and (2) sub-functionalization depends on cell type and experimental conditions, including extent of knockdown and compensation. More research will be required to ascertain the sub-specialization of the two cohesin complexes in centromeric cohesion. We speculate that the low enrichment of cohesin within CenHaps is due to the absence of CTCF binding sites, low levels of transcription and low gene density in these regions, in comparison to the rest of the genome, as these factors are associated with stable cohesin binding[33]. Furthermore, stretching and recoil of centromeric α-satellite DNA in response to microtubule activity could also contribute to cohesin instability within these regions.

Our evidence from super-resolution microscopy experiments shows that cohesin is arranged on mitotic chromosomes in an axis between the sister chromatids (Figs. 2 and 3). Cohesin complexes can interact with DNA in topological and non-topological conformations[45]. There are two forms of DNA entrapment by cohesin in the genome – the cohesive form between sisters and the loop extruding form entrapping a single chromatid[46,47]. While much of the cohesive cohesin is lost from chromosome arms beginning at prometaphase[48–50], it is retained at pericentromeric sites between sister chromatids[51]. We predict that both cohesive and loop extruding forms of cohesin[52] shape human centromere organization. We speculate that the enrichment of cohesin along the vertical axis of the centromere represents the cohesive form of cohesin entrapping sister chromatids at pericentromeric regions. In addition to cohesin-mediated cohesion, sister chromatids are held together at the centromeres by at least one other mechanism – catenations[53]. Our ChIP and microscopy experiments suggest that the active α-satellite DNA does not participate in sister chromatid cohesion, unless it is via catenations. Additional proteins like condensin[16,21], topoisomerase IIA[54,55] and CENP-B[56], likely participate in shaping the core structure of human centromeres, and our methods can be extended to determine their placement.

From our SIM results (Fig. 5a, zoom and Fig. 6a, zoom), we envisage that α-satellite DNA forms a primary extensible loop with CENP-A-enriched chromatin at the tip of this loop, and loop extruding cohesin at the base (Fig. 7). This arrangement allows for the stretching of the α-satellite DNA by microtubules. Further, we and others[21] variably detect a small fraction of cohesin immediately exterior to the α-satellite DNA domain (Fig. 3b, d). We speculate that this cohesin is the loop extruding form and might be loosely associated with the CENP-A enriched chromatin, or only associated during mitosis, and hence not detected as significant peaks in our ChIP-seq assay (Fig. 1c). A singular loop of DNA would not be sufficient to withstand microtubules forces[14], nor would it be visualized with our FISH probes. We speculate that similar to the bottlebrush configuration composed of centromeric DNA loops from all 16 chromosomes proposed in budding yeast[57], several additional loops of α-satellite DNA exist along the primary extensible loop, giving it the appearance of a DNA cloud along the axis of the chromosome (Fig. 6b, Fig. 7). We speculate that the length of the α-satellite DNA array would dictate the size of these loops/folds and thereby the spread of this DNA along the chromosomal axis (Fig. 6e). This DNA may be organized by additional structural proteins not analyzed in our study.

Human α-satellite arrays are highly polymorphic in sequence and size between chromosomes and also in the human population[12,58]. However, the process of mitosis is not just carefully controlled and conserved across every human chromosome but is evolutionarily conserved across all metazoans. This begs the question of how sequence and size polymorphisms attain a common core centromere geometry to support mitosis. The consistent inter-CENP-A distances across many chromosomes indicates that this geometry can be achieved despite huge variation in the size of α-satellite arrays. A similar core structure is likely critical for centromere function on the mitotic spindle. Our data strongly suggests that flexible organization of α-satellite DNA allows a similar core structure to form on very differently sized HOR arrays. The location of sister chromatid cohesion at pericentromeres away from α-satellite DNA lessens the possibility that cohesion varies with array size. In experiments examining the position of RAD21, we identified central, uniform and split subcategories. Although we do not know the origins of these classes, one possibility could be how much α-satellite DNA is being packaged. For example, the split category could accomodate a large α-satellite DNA array. Investigating these structures on other chromosomes with differently sized α-satellite arrays will help resolve this question in the future.

Our model suggests that human centromeres, while abundantly polymorphic, assume a similar core geometry under native conditions. We find evidence that large differences in α-satellite array size can be accommodated in a common core geometry. We also find that the two functional domains of centromeres – the cohesive domain and the kinetochore microtubule attachment domains are spatially separated in both linear and 3 dimensions on mitotic chromosomes. The location of cohesion in pericentromeric regions away from the kinetochore may be a unifying principle of point and regional centromere structure. Due to the highly repetitive nature of human centromeres, they are refractory to examination by current short-read based Hi-C techniques to deduce the 3D organization of these important regions of the genome. The methods we have developed will facilitate future work layering in additional factors and conditions to obtain a holistic view of centromere organization and function.

## Methods

### Cell culture

hTERT RPE-1 cells were sourced from ATCC (CRL-4000) and cultured in DMEM:F12 medium (Gibco) supplemented with 10% FBS (Peak Serum Inc.) and 1X Glutamax (Gibco). hTERT CHM13 cells[22] obtained from the T2T consortium were adapted and cultured in DMEM:F12 medium (Gibco) supplemented with 10% FBS, 1X Glutamax (Gibco), 1X Insulin-transferrin-selenium (Gibco), 1X sodium pyruvate (Gibco). hTERT RPE-1 cells with endogenous SMC1A locus tagged at the C-terminus with mEGFP were generated using CRISPR-Cas9-based genome editing. crRNA sequence used for this was AAAATACTG CTACTGCTCAT. SMC1A-mEGFP homology arm donor plasmids deposited by Allen Institute were obtained from Addgene (Plasmid #114406). All cells were cultured at 37 °C, 5% $CO_2$ under humidity-controlled conditions. Cells were routinely tested for absence of *Mycoplasma* contamination using PCR.

### Chromatin immunoprecipitation (ChIP)

Two independent ChIP replicates were performed for cohesin subunits RAD21, SA1 and SA2 in hTERT RPE-1 (hereafter RPE-1) and three independent replicates of ChIP were performed in hTERT CHM13 (hereafter CHM13) cell lines. CHM13 ChIP-seq experiments were performed in presence of 1% *Drosophila* Kc167 cell chromatin as an internal control. Antibodies used were as follows—rabbit anti-RAD21 (Abcam, ab154769) for RPE-1, rabbit anti-RAD21 (Abcam, ab992) for CHM13, rabbit anti-SA1 (Bethyl, A302-579A), rabbit anti-SA2 (Bethyl, A302-580A). RPE-1 and CHM13 cells were cultured in 150 mm dishes

(Corning) up to 80% confluence in 20 ml of culture medium. To fix cells, 16% methanol-free formaldehyde solution (Pierce) was added directly to the culture medium to a final concentration of 1% and incubated at room temperature for 10 min with gentle agitation. Formaldehyde was quenched with glycine (final = 125 mM) for 5 min at room temperature. Cells were washed 3 times in 1x ice-cold phosphate buffered saline (PBS) containing 1 mM PMSF, 1x protease inhibitory cocktail (PIC) (Roche) and harvested by using a cell lifter, spun down at 425 g for 15 min (RPE-1)/153 g for 8 min (CHM13) at 4 °C. Pellet was washed in 2 ml Buffer A2 (15 mM) HEPES, 140 mM NaCl, 1 mM EDTA, 0.5 mM EGTA, 1% Triton X-100, 0.1% sodium deoxycholate, 1% SDS, 0.5% N-lauroylsarcosine, 1 mM DTT, 1 mM PMSF, 1x PIC, spun at 425 g/1800 rpm 344 g for 5/8 min at 4 °C (RPE-1 and CHM13 respectively). For RPE-1 cells, $8 \times 10^6$ cells were re-suspended in 1.8 mL Buffer A2 and distributed in 6 nos. 1.5 mL Bioruptor® tubes. Cells were sonicated using Bioruptor bath sonicator for 2 rounds of 7 cycles (1 cycle: 30" ON, 30" OFF) at 4 °C with 1 min of incubation on ice followed by vortexing between each round. $1 \times 10^7$ CHM13 cells, were resuspended in 1 mL of buffer A2 and aliquoted into 1.5 mL tubes (Covaris®). Cells were sonicated for 8 min (Power: 140, Duty factor: 6, 200 bursts/cycle) on Covaris S220 ultrasonicator. Sonicated chromatin was pooled and spun at 20,817 g for 15 min, 4 °C and the pellet was discarded.

All steps from here on were performed in Low protein binding microcentrifuge tubes (Thermo Scientific, 90410). Bead preparation and antibody binding: 30-50 μL of Dynabeads™ Protein G (Invitrogen, 10003D) were used per ChIP. Beads for each ChIP were distributed in 1.5 mL tubes, washed twice in 1x PBS and once in 0.5% bovine serum albumin (BSA)-1x PBS. For antibody binding, each tube of beads was resuspended in 800 μL of 0.5% BSA-1x PBS and incubated with 5 μg of each antibody for 3 h at 4 °C on a nutator at low speed. Antibody bound beads were washed once in 0.5% BSA-1x PBS. Pre-clearing: Dynabeads™ Protein G beads were prepared as before and 10 μL of blocked beads were used to pre-clear ~100 μg of chromatin for 3 h at 4 °C. 10% of pre-cleared chromatin was removed as Input and flash frozen in liquid nitrogen. Chromatin immunoprecipitation: ~75–100 μg of pre-cleared chromatin (amounts varied across replicates but constant for each antigen in a single experiment) was added to antibody-bead conjugates, volume was made up to 800 μL with Buffer A2 and incubated overnight at 4 °C on a nutator at low speed. The following day, beads were collected using Dynamag™ (Invitrogen) and supernatant was discarded. Beads were washed 3 times in low salt buffer (20 mM Tris-HCl pH 8.0, 150 mM NaCl, 2 mM EDTA, 1% Triton X-100), 3 times in high salt buffer (20 mM Tris-HCl pH 8.0, 500 mM NaCl, 2 mM EDTA, 1% Triton X-100) and once in LiCl buffer (20 mM Tris-HCl pH 8.0, 250 mM LiCl, 1 mM EDTA, 1% Triton X-100, 0.1% NP-40, 0.5% sodium deoxycholate). Washes were 10 min each at 4 °C. Chromatin elution and de-cross-linking: 100 μL elution buffer (EB) (50 mM Tris-HCl pH 8.0, 10 mM EDTA, 1% SDS) and 100 μL TE (10 mM Tris-HCl pH 8.0, 1 mM EDTA) were added to beads in each tube and incubated at 65 °C for 15 min. Supernatant was collected and the step was repeated once. Volume of Input was made up to 400 μL with 1:1 EB:TE and treated similarly. NaCl (final = 210 mM) and 20 μg (2 μL) Proteinase K (Invitrogen) were added to each tube of eluted chromatin overnight at 65 °C on a thermomixer at 600 rpm. 10 μg of RNaseA (Invitrogen) was added to each tube and incubated at 37 °C for 1 h. DNA purification: DNA was extracted with equal volume of phenol:chloroform:isoamyl alcohol (25:24:1) and the aqueous layer was transferred to new tube containing NaCl (final = 200 mM) and 2 μL GlycoBlue™ (ThermoFisher Scientific, AM9515). 750 μL of ice-cold 100% ethanol was added, and tubes were incubated at −80 °C for 1 h, centrifuged at 20,817 g for 30 min at 4 °C, DNA pellet was washed once with 70% ethanol and air-dried. Finally, DNA pellets were re-suspended in ultrapure distilled water (Invitrogen).

## Sequencing and analysis

Libraries were prepared using the KAPA HTP Library Prep Kit for Illumina (Roche, KK8234) and Bioo Scientific NEXTflex DNA barcodes (Perkin Elmer, NOVA-514104). The resulting libraries were purified using the Agencourt AMPure XP system (Beckman Coulter, A63882) then quantified using a Bioanalyzer HS DNA kit (Agilent Technologies, 5067-4626) and a Qubit fluorometer (Life Technologies) with Qubit dsDNA HS assay kit (Invitrogen, Q32851). Post amplification size selection (275–700 bp) was performed on all libraries using a PippinHT (Sage Science) using 1.5% cassette (HTC1510). Libraries were pooled, requantified, and sequenced as 150 bp paired reads on a mid-output flow cell using the Illumina NextSeq 500 instrument. Following sequencing, Illumina Real Time Analysis version RTA 2.4.11 and bcl2fastq2 v2.20 were run to demultiplex reads and generate FASTQ files.

## ChIP-sequencing analysis

ChIP-seq data was aligned to CHM13 v1.0 reference genome using bwa version 0.7.17-r1188 with the following parameters: bwa mem -k 50 -c 1000,000. Secondary and supplementary alignments were excluded using samtools version 1.14 to filter out reads with SAM FLAG 2308. Read coverage was calculated and normalized to Reads Per Million from BAM files using the GenomicRanges version 1.48.0 library in R. Peaks were called using macs version 2.1.2 with the following parameters: -g 2.9e9 -q 0.01. Reference peak sets for each factor were created using the reduce function from GenomicRanges to create a single collection of candidate loci, and then discarding any loci not observed in at least 2 replicates. CTCF ChIP-seq data[59] obtained from GEO with accession number GSE30263, for samples GSM749673, GSM749771, GSM1022665, representing two CTCF IPs and one total chromatin input, was aligned to the CHM13 genome using bowtie2 version 2.4.2 with default parameters. Coverage in reads per million and a ratio track of IP over total chromatin in log2 format was created using GenomicRanges. The ratio was moderated by adding 1 to the numerator and denominator to prevent division by zero. Two samples of paired-end RNA Seq data from[60] were aligned to the CHM13 genome using STAR 2.7.3a with default parameters along with chm13.draft_v1.0 gene annotation from the T2T consortium to generate a count table. Gene counts were converted to RPKM and averaged for abundance analysis.

## Metaphase chromosome preparation

RPE-1 cells plated in 100 mm dishes were treated with Colcemid (100 µg/mL) for 4 h. Cells were washed once with ice-cold DPBS and mitotic cells were harvested by shake-off in ice-cold DPBS. Cells were resuspended in 75 mM KCl at a density of $3 \times 10^5$ cells/ml for 10 min. Approximately $3 \times 10^4$ cells were spun onto ethanol washed 24 × 50 mm glass coverslips (Cat# 16004-322) at 366 g for 5 min using Thermo scientific Shandon Cytospin 4. Coverslips were incubated in KCM buffer (120 mM KCl, 20 mM NaCl, 10 mM Tris-HCl, ph8, 0.5 mM EDTA, 0.1% (v/v) Triton X-100) for 15 min and fixed in 4% (v/v) paraformaldehyde (PFA)/KCM for 8 min, washed and stored at 4 °C until further use. Coverslips were blocked in 5% (w/v) BSA/KCM for 60 min and incubated with primary antibodies diluted in KCM buffer overnight at 4 °C. Antibodies used are as follows - rabbit anti-RAD21 (Abcam, ab154769, dilution 1:100), mouse anti-CENPA (MBL, D115-3, dilution 1:200), rabbit anti-CENPB (Abcam, ab25734, dilution 1:200). SMC1A-mEGFP on metaphase spreads was probed using an anti-GFP antibody (Abcam, ab6556, dilution 1:100), as endogenous molecules of SMC1A-mEGFP were not enough to be directly detected without signal amplification by secondary antibodies. Coverslips were washed in KCM buffer and incubated with secondary antibodies for 1 h and washed. Secondary antibodies used – Donkey anti-rabbit CF®568 (Biotium, Cat# 20098, dilution 1:300), Donkey anti-Rabbit CF®647 (Biotium, Cat# 20047, dilution 1:300), Donkey anti-mouse

(Biotium, Cat# CF®488 A, dilution 1:300). Metaphase spreads were counterstained with DAPI and mounted in Prolong Glass antifade (Thermo Fisher, Cat# P36980). Prolong glass is a hardening medium with a refractive index of 1.52 and is most compatible with the refractive index of the glass coverslips 1.525 used in this study. Prolong glass has a minor flattening effect on chromosome spreads. However, the use of these thin coverslips and the corresponding hardening mounting medium is crucial to obtain the best resolution for Structured Illumination Microscopy. Thus, given the trade-off between improved resolution or flattening of chromosomes, Prolong Glass medium was the appropriate choice for this imaging methodology.

## Immuno-FISH on metaphase chromosomes

After Immunofluorescence staining, samples were fixed with 4% paraformaldehyde for 10 min, rinsed with PBS two times, and dehydrated in ethanol series (70, 80, 100%). Probes used for centromere 7 (D7Z1) FISH was a Satellite Enumeration Probe from Cytocell (Cat# LPE 007G-A) and for CENP-B box FISH was a Cy3-labeled PNA probe from PNA Bio with the sequence 5′-ATTCGTTGGAAACGGGA-3′. Hybridization buffer used was from Empire Genomics. Hybridization mixture was applied on samples, covered with a small piece of coverslip, and sealed with Cytobond (SciGene cat#2020-00-1). Samples were denatured on a heating block at 75 °C for 3 min, and hybridization was performed at 37 °C for 16 to 20 h. Samples were washed with 2X SSC for 5 min, 50% formamide/2X SSC for 15 min at 37 °C, 2X SSC for 10 min two times, and 2X SSC/0.1% Triton X for 10 min. Samples were stained with secondary antibodies and 10 ug/ml of DAPI for 60 min at RT and washed with 2X SSC three times. ProLong Glass antifade mountant (Invitrogen, Cat# P36980) was applied and samples were cured overnight.

## Confocal microscopy

Cells were imaged with Zen software (black edition, version 14.0.24.201) on a Zeiss LSM780 confocal microscope with 405-nm, 488-nm, and 561-nm laser lines using a 63× Plan-Apochromat 1.4-numerical-aperture (NA) oil immersion objective at 2.5x digital zoom. Scanning was performed sequentially (x-y, 512 pixels by 512 pixels [1 pixel ~ 0.105 µm]), and z-stacks were collected at a step size of 0.34 µm and a pinhole size of ~0.7 µm (1 AU). The pixel depth was 16 bits, the line averaging was 4, and the scan speed was 10.

## 3D-Structured illumination microscopy (3D-SIM) data acquisition and reconstruction

3D-structured illumination microscopy (3D-SIM), quality checks and reconstruction were performed as previously described[61,62]. Paraformaldehyde-fixed and immunostained metaphase chromosomes were imaged using the Applied Precision OMX Blaze V4 structured illumination microscope (GE Healthcare) equipped with a 60x 1.42 NA Olympus Plan Apo oil objective and three PCO Edge sCMOS cameras. Refractive index of oil was adjusted appropriately for each experiment (Range: 1.514 – 1.526). Two-color (Alexa 488/Alexa 568) imaging was performed using 488 or 561 nm lasers, respectively with alternating excitation and a 405/488/561/640 dichroic with 504–552 and 590–628 nm emission filters. DAPI channels were imaged in widefield mode using 405 nm with the same dichroic and 420–451 nm emission filters. z-stacks were collected at a step size of 125 nm to cover the thickness of the centromeres (typically 700 nm) and a few stacks above and below this. SIM images were reconstructed using SoftWoRx software version 6.5.2 (GE Healthcare) with a Wiener filter of 0.001. Images for co-labeled CENP-A and D7Z1 or CENP-B FISH probes, and SMC1A-mEGFP were obtained using the Lattice SIM technology on the Elyra 7 microscope (Carl Zeiss AG) with a similar setup. The acquisition was done using a 63x oil immersion objective lens (Plan-Apochromat 63x/1.40 Oil), the

illumination pattern was set to 15 phases, and the z-stack spacing was set at 100 nm with a similar range. The green and red channels had slightly different emission ranges on the Elyra 7 (495–550 nm for green and 570–620 nm for red). The raw SIM images were processed using the ZEN software (from ZEISS) with manual adjustments for sharpness in the range of 10-11.

### 3D-SIM image analysis

For all analyses sum projections of individual channels were used. Pairs of CENP-A clusters on sister chromatids were either hand annotated or found automatically using peak finding in python. Once identified, 3-dimensional Gaussians were fitted to the CENP-A clusters to fine tune their position. Individual pairs were then rotated about the midpoint of the two peaks to be vertically aligned and consistently cropped and centered. From the rotated and centered images, the slice corresponding to the center of the structure was extracted and kymographs and averaged images generated. As the orientation of the chromosomes were random, kymographs were generated from combining both original and mirrored versions of these images and then integrating signal within 5 pixels of the center line in the lateral (profile perpendicular to chromosome) and vertical (profile parallel to chromosome) directions. This corresponds to a width either of 155 nm (Elyra7) or 200 nm (OMX Blaze). For determining the width of CENP-A in the vertical direction a larger integration zone of 41 pixels was used to encompass the two peaks off the chromosomal axis.

To investigate the chromosome-to-chromosome variability of the distribution of RAD21 and Centromere 7, we found the ratio of the signal intensity from 160 – 320 nm off center to the signal intensity within 120 nm of the center for each kymograph. This ratio was then used to sort the chromosomes from edge-enriched to center-enriched. We partitioned these into three equal classes and then summed together the image sets from each class to look at their spatial localizations.

Gaussian peaks were fitted using scipy's optimize module and Aikaike Information Criterion was used to decide whether to use two or three peaks when ambiguous.

### Statistical analysis

Statistical analysis was performed using R or GraphPad Prism v 9.0. Statistical tests used for each analysis have been mentioned in respective figure legends.

### Reporting summary

Further information on research design is available in the Nature Portfolio Reporting Summary linked to this article.

## Data availability

The raw ChIP-sequencing data generated from this study have been deposited to the GEO database under the accession code - GSE240957. Original data underlying this manuscript can be accessed from the Stowers Original Data Repository (ODR) at http://www.stowers.org/research/publications/libpb-2418. Access to the Stowers ODR is available to the public and generally free. Occasionally, access to very large data files for which internet download is not practical may require special arrangements and may include a fee to cover the cost of storage media or other method of transfer. Source data are provided with this paper.

## Code availability

3D-SIM images obtained after reconstruction were analyzed using several custom open-source FIJI analysis packages available at http://research.stowers.org/imagejplugins/ and custom codes available at https://zenodo.org/record/8427422.

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

## Acknowledgements

We thank members of the Gerton lab for discussion and feedback on this work. Research reported in this publication was supported by the Stowers Institute for Medical Research and NIH-NCI under award number R01CA266339 (to JL Gerton). We thank Mark Miller for help with illustrations.

## Author contributions

A.S.G.—conceptualization, experimentation, data acquisition, formal analysis, validation, investigation, visualization, methodology, and writing. C.S.—software, formal analysis, visualization, methodology. D.T.—experimentation. S.M.K.—software, formal analysis, visualization, methodology, and writing. Z.Y.—data acquisition, software. S.S.—methodology. J.U.—methodology. J.L.G.—conceptualization, writing, funding acquisition.

## Competing interests

The authors declare no competing interests.
