## [Peer Review File · Nature Communications]

Defining a core configuration for human centromeres during mitosisREVIEWER COMMENTS

Reviewer #1 (Remarks to the Author):

Review of Sen Gupta et al., Defining a core configuration for human centromeres during mitosis.

This is a study of the distribution of CENP-A and cohesin within the human centromere. The authors use super-resolution microscopy of fixed cells, together with ChIP analysis and the newly available assemblies of centromere DNA to deduce the higher order structure of the human centromere.

The first conclusion is that the sites of enriched CENP-A are depleted of cohesin. Cohesin is associated preferentially with the pericentromere. This recapitulates several studies in budding yeast, demonstrating the stratification of Cse4 from cohesin (reviewed in Mcainsh, AD & Marston, AL 2022, 'The four causes: The functional architecture of centromeres and kinetochores', Annual review of genetics, vol. 56, no. 1, pp. 279-314. <https://doi.org/10.1146/annurev-genet-072820-034559>; Tripartite Chromatin Localization of Budding Yeast Shugoshin Involves Higher-Ordered Architecture of Mitotic Chromosomes. Deng X, Kuo MH. G3 (Bethesda). 2018 Aug 30;8(9):2901-2911. doi: 10.1534/g3.118.200522; Shaping centromeres to resist mitotic spindle forces. Lawrimore J, Bloom K. J Cell Sci. 2022 Feb 15;135(4):jcs259532. doi: 10.1242/jcs.259532. Epub 2022 Feb 18.).

The higher order organization of cohesin was difficult to understand. As in yeast, the bolus of cohesin is between the sister chromatids (localized to the pericentromere) and not overlapping with CENP-A (Cse4 in yeast), that is microtubule proximal, distal from the central sister chromatid axis. The authors show there is a depletion zone of cohesin in the center of the structure (Fig 3E). In budding yeast, this depletion zone has been shown through model convolution to reflect the barrel-like organization of cohesin. Is the cohesin organized as a barrel that is aligned along the sister chromatid axis and perpendicular to the kt mt axis.

The issue of cohesin organization is germane to the classification of cohesin into three classes (Fig. 4). With higher order structures such as cohesin, the tilt of the specimen will give rise to very different images. In a study from Stephens et al., (The spatial segregation of pericentric cohesin and condensin in the mitotic spindle. Stephens AD, Quammen CW, Chang B, Haase J, Taylor RM 2nd, Bloom K. Mol Biol Cell. 2013 Dec;24(24):3909-19. doi: 10.1091/mbc.E13-06-0325. Epub 2013 Oct 23.), the authors simulated microscope images of cohesin from different angles (Fig. 2). My question is whether the different classes herein are simply different perspectives of one unifying structure. Are there any fiducials the authors can use to estimate the tilt of the specimens under study.

The discussion would benefit from acknowledging prior studies that come to very similar conclusions. It is stated that the work solves a long-standing conundrum in the field as to the role of cohesin in supporting both cohesin and stretching. This conundrum has been addressed in the work from budding yeast that demonstrated the critical role of a bottlebrush of chromatid loops in the pericentromere serving as a mechanism to build tension. In fact, Fig. 7. Is remarkably similar to several figures from Lawrimore (Fig. 4 J Cell Sci. 2022 Feb 15;135(4):jcs259532.doi: 10.1242/jcs.259532. Epub 2022 Feb 18.; Fig. 6 J Cell Biol. 2015 Aug 17;210(4):553-64.doi: 10.1083/jcb.201502046.).

The authors discuss the differential functions of cohesin in the pericentromere vs the arms (p 13, lines 358-374). The differential functions of cohesin was addressed in budding yeast from the work of Koshland (Curr Biol. 2010 May 25;20(10):957-63.doi:

10.1016/j.cub.2010.04.018. Epub 2010 May 6). In that work, the authors used quantized reduction of cohesin to demonstrate that the pericentric cohesin was refractory to reduction and reflected a critical role for pericentric cohesin, similar to being discussed in this manuscript.

The extensible loops of alpha-satellite are structurally similar to the bottlebrush configuration proposed in Stephens and Lawrimore (ibid.). That this arrangement allows for DNA stretching by microtubules (line 378, p. 13) is precisely that proposed in these earlier studies.

In summary, this is a molecular and microscopic analysis of the localization of cohesin, alpha-satellite DNA and CENP-A. The data are fairly clear, but in several instances are difficult to deduce the underlying structure. The conclusions are similar to those proposed in budding yeast for both the specific role of cohesin in the pericentromere and the lateral organization of loops. I suggest the authors refer to prior studies to better reflect the contributions herein.

Reviewer #2 (Remarks to the Author):

The manuscript entitled "Defining a core configuration for human centromeres during mitosis" by Gupta et al. from the lab of Jennifer Gerton is definitely suitable for publication in Nature Communications. There are a few critical controls that need to be produced to manifest the findings of the paper and rule out any over-interpretation. Overall however, the authors make the exciting observation that the two key centromere functions of building kinetochores and maintaining cohesion in mitosis are spatially separated. They achieve this by using state-of-the art quantitative 3D-Structured Illumination Microscopy, which is still in its infancy. The authors combine super-resolution microscopy with calibrated ChIP and FISH to study the spatial organisation of centromeres, specifically CenpA and cohesin proteins in a variety of cell lines and conclude that while CenpA localises to a-satellites, cohesin is enriched at the pericentric regions.

Main point

I do not have any main conceptual points. The paper is well written and coherent. The findings are important and exciting. The established workflow has potential to be used for further studies.

My main concerns are of technical nature.

First of all, the ChIP experiments are controlled by re-alignment to previously published data and variety of used cell lines. All ChIP experiments are however carried out with antibodies against target proteins, which notoriously show unspecific binding. This does not matter to localise the proteins to chromatin, but as soon as localisations of different proteins are compared, ChIP needs to be calibrated by spike-in of a different species genome. This is an incredibly easy and effective step.

If the authors want to conclude only that CENPA and cohesin show exclusive localisations on chromatin, the methodology used can remain. The paragraph about overlap discrepancies especially in CHM13 cells needs to either be removed without calibration as it is most likely coming from missing calibration. If overlap of peaks should be discussed by authors, please repeat the ChIP with calibration. It is impossible to discuss overlap frequencies without calibration.

The 3D SIM is carried out very well. The technique, imaging conditions and analysis are sound and I do not admire the authors for the amount of work it must have been to quantitate all this data. It is important to note that this is by no means routine and that it presents a huge benefit for the presented data.

My main concern with the SIM samples is that they have been mounted in prolong glass, which completely destroys 3D chromatin architecture by flattening the chromosomes entirely (according to our own experiments and rigorous comparison of mounting media for 3D-SIM) This makes this more likely to be 2D-SIM and does not allow accurate 3D measurements.

I have to ask the authors to remake the key samples in Slowfade or another suitable non hardening mounting media and reproduce the key observation at least for the supplementary data. I do not ask for extensive quantification, and the spatial exclusion will most likely not change but the measurements need to be in the same ballpark otherwise the exact numbers might be incorrect and mis-lead the research community.

My second experimental request is that the authors produce controls to prove 3D-SIM resolution in axial and lateral dimension as well as all channels.

I disagree that the fact that cohesin is measured to 83 nm makes it likely to be two cohesin complexes. The standard resolution of 3D-SIM is >100 nm and therefore conclusions like these cannot be supported. Furthermore within one channel in 3D-SIM a single cohesin ring with primary and secondary antibody will like be around that size. In addition, conclusions about whether loops are tethered or sisters are encircled should be removed as they are mere speculation given that the molecular mechanisms of either process are still a mystery.

My final concern is that antibodies have been used to obtain all 3D-SIM data. We need to see proof in this study that the antibodies are specific (IF after siRNA or degran) and that they do not exclude each other sterically, which would then produce the observed result. I agree, it is unlikely, but very easy to control, by co-localising RAD21 with SMC1 or similar.

Finally, the model needs work. It is incredibly messy and unhelpful as the lines are very thick, hard to read annotations half hidden behind drawings.

Minor points

I would not call this approach SPA-SIM. SIM is a microscopy technique and pre-fixes should reflect improvements in the technology. Single particle averaging is a standard procedure used in imaging and an analysis tool. As described above, it is done excellently but does not need to be hyped by generating yet another acronym.

The paper Woglar et al, 2020 from Anne Villeneuve's lab needs to be cited as this was the first time SIM of a SMC protein on chromosomes was published and it is worth noting and in agreement with the here shown data that Rec8 also localises to the inter-axial space in meiosis.

It would be helpful at the transition from confocal to 3D-SIM to mention that SIM has an 8-fold volumetric resolution increase over confocal microscopy.

In conclusion, I fully support this paper for publication after inclusion of the above-mentioned controls and if they are in line with the currently presented data. The

paper is well written, provides a significant advancement in our understanding of mitosis and has potential to be of greater impact to the community by now allowing studies of centromeric cohesion in the 3D chromatin space. Well done!

We thank both reviewers for their insightful comments and have incorporated revisions in the text as well as additional experiments to address their concerns. In summary we have

- 1) added more references to contextualize our study better within the existing literature, especially with regard to budding yeast models
- 2) added data to address imaging methods and limitations, as two figures for the reviewers
- 3) added a new side-on view of Rad21 signal (Figure 4c)
- 4) added imaging data from an Smc1a-GFP cell line that replicates the findings for Rad21 (Figure S2)
- 5) removed comparisons with publicly available cohesin datasets due to their lack of spike in controls (Figure 1 and S1)
- 6) improved the model figure (Figure 7)

REVIEWER COMMENTS

Reviewer #1 (Remarks to the Author):

Review of Sen Gupta et al., Defining a core configuration for human centromeres during mitosis.

This is a study of the distribution of CENP-A and cohesin within the human centromere. The authors use super-resolution microscopy of fixed cells, together with ChiP analysis and the newly available assemblies of centromere DNA to deduce the higher order structure of the human centromere.

The first conclusion is that the sites of enriched CENP-A are depleted of cohesin. Cohesin is associated preferentially with the pericentromere. This recapitulates several studies in budding yeast, demonstrating the stratification of Cse4 from cohesin (reviewed in Mcainsh, AD & Marston, AL 2022, 'The four causes: The functional architecture of centromeres and kinetochores', Annual review of genetics, vol. 56, no. 1, pp. 279-314. <https://doi.org/10.1146/annurev-genet-072820-034559>; Tripartite Chromatin Localization of Budding Yeast Shugoshin Involves Higher-Ordered Architecture of Mitotic Chromosomes. Deng X, Kuo MH. G3 (Bethesda). 2018 Aug 30;8(9):2901-2911. doi: 10.1534/g3.118.200522; Shaping centromeres to resist mitotic spindle forces. Lawrimore J, Bloom K. J Cell Sci. 2022 Feb 15;135(4):jcs259532. doi: 10.1242/jcs.259532. Epub 2022 Feb 18.).

The higher order organization of cohesin was difficult to understand. As in yeast, the bolus of cohesin is between the sister chromatids (localized to the pericentromere) and not overlapping with CENP-A (Cse4 in yeast), that is microtubule proximal, distal from the central sister chromatid axis. The authors show there is a depletion zone of cohesin in the center of the structure (Fig 3e). In budding yeast, this depletion zone has been shown through model convolution to reflect the barrel-like organization of cohesin. Is the cohesin organized as a barrel that is aligned along the sister chromatid axis and perpendicular to the kt mt axis.

Response –

The barrel model from yeast represents multiple centromeres on a single spindle, which have been proposed to be analogous to a single human chromosome. Our data, derived from single resting human metaphase chromosomes in the absence of a spindle, do not show evidence of

cohesin barrels, even in side views of chromosomes (Fig. 4C). Cohesin may be dynamically redistributed at the centromeres when chromosomes are on the spindle, or may show different localization when chromosomes are not flat on a slide, which will be part of our future efforts.

The issue of cohesin organization is germane to the classification of cohesin into three classes (Fig. 4). With higher order structures such as cohesin, the tilt of the specimen will give rise to very different images. In a study from Stephens et al., (The spatial segregation of pericentric cohesin and condensin in the mitotic spindle. Stephens AD, Quammen CW, Chang B, Haase J, Taylor RM 2nd, Bloom K. *Mol Biol Cell*. 2013 Dec;24(24):3909-19. doi: 10.1091/mbc.E13-06-0325. Epub 2013 Oct 23.), the authors simulated microscope images of cohesin from different angles (Fig. 2). My question is whether the different classes herein are simply different perspectives of one unifying structure. Are there any fiducials the authors can use to estimate the tilt of the specimens under study.

Response –

We completely agree with the reviewer that different tilts of specimen may give rise to apparent structural differences. In fact, since our metaphase chromosomes are fixed in paraformaldehyde (not the more flattening methanol fixation) to maintain structures as close to 3D as possible, we did observe many chromosomes that did not lie flat on the coverslip surface. To circumvent this issue, we supplemented our automated CENP-A peak finding methodology with careful manual inspection to ensure that chromosomes used in the analysis displaying different cohesin classes were lying flat on the coverslip and not bent or tilted. Furthermore, a tilted sample would result in extreme differences in inter-CENP-A foci distances between the cohesin classes. However, our analysis shows that the inter-CENP-A foci distances are remarkably conserved between different cohesin classes (Figure 4e), suggesting that the different cohesin classes are not an outcome of sample tilt.

The discussion would benefit from acknowledging prior studies that come to very similar conclusions. It is stated that the work solves a long-standing conundrum in the field as to the role of cohesin in supporting both cohesin and stretching. This conundrum has been addressed in the work from budding yeast that demonstrated the critical role of a bottlebrush of chromatid loops in the pericentromere serving as a mechanism to build tension. In fact, Fig. 7. Is remarkably similar to several figures from Lawrimore (Fig. 4 *J Cell Sci*. 2022 Feb 15;135(4):jcs259532.doi: 10.1242/jcs.259532. Epub 2022 Feb 18.; Fig. 6 *J Cell Biol*. 2015 Aug 17;210(4):553-64.doi: 10.1083/jcb.201502046.).

Response –

We now cite and discuss previous studies that have addressed the organization of budding yeast point centromeres. We point out the similarities between budding yeast and our proposed model of human centromeres, the strengths and limitations of previous work, and clarify that the data helps resolve the conundrum for human centromeres.

The authors discuss the differential functions of cohesin in the pericentromere vs the arms (p 13, lines 358-374). The differential functions of cohesin was addressed in budding yeast from the

work of Koshland (Curr Biol. 2010 May 25;20(10):957-63.doi: 10.1016/j.cub.2010.04.018. Epub 2010 May 6). In that work, the authors used quantized reduction of cohesin to demonstrate that the pericentric cohesin was refractory to reduction and reflected a critical role for pericentric cohesin, similar to being discussed in this manuscript.

Response –

We are aware of this work and its implications. However, since we did not carry out knockdown experiments, it was challenging to incorporate the quantized reduction results from budding yeast. Future knockdown studies will be interpreted in light of this previous work.

The extensible loops of alpha-satellite are structurally similar to the bottlebrush configuration proposed in Stephens and Lawrimore (ibid.). That this arrangement allows for DNA stretching by microtubules (line 378, p. 13) is precisely that proposed in these earlier studies. In summary, this is a molecular and microscopic analysis of the localization of cohesin, alpha-satellite DNA and CENP-A. The data are fairly clear, but in several instances are difficult to deduce the underlying structure. The conclusions are similar to those proposed in budding yeast for both the specific role of cohesin in the pericentromere and the lateral organization of loops. I suggest the authors refer to prior studies to better reflect the contributions herein.

Response –

We have expanded the literature review to discuss relevant literature from yeast models of centromeres.

Reviewer #2 (Remarks to the Author):

The manuscript entitled “Defining a core configuration for human centromeres during mitosis” by Gupta et al. from the lab of Jennifer Gerton is definitely suitable for publication in Nature Communications. There are a few critical controls that need to be produced to manifest the findings of the paper and rule out any over-interpretation. Overall however, the authors make the exciting observation that the two key centromere functions of building kinetochores and maintaining cohesion in mitosis are spatially separated. They achieve this by using state-of-the-art quantitative 3D-Structured Illumination Microscopy, which is still in its infancy. The authors combine super-resolution microscopy with calibrated ChIP and FISH to study the spatial localization of centromeres, specifically CenpA and cohesin proteins in a variety of cell lines and conclude that while CenpA localizes to α -satellites, cohesin is enriched at the pericentric regions.

Main point

I do not have any main conceptual points. The paper is well written and coherent. The findings are important and exciting. The established workflow has potential to be used for further studies.

Response –

We are glad to hear this positive assessment of our manuscript.

My main concerns are of technical nature.

First of all, the ChIP experiments are controlled by re-alignment to previously published data and variety of used cell lines. All ChIP experiments are however carried out with antibodies against target proteins, which notoriously show unspecific binding. This does not matter to localize the proteins to chromatin, but as soon as localisations of different proteins are compared, ChIP needs to be calibrated by spike-in of a different species genome. This is an incredibly easy and effective step.

If the authors want to conclude only that CENPA and cohesin show exclusive localisations on chromatin, the methodology used can remain. The paragraph about overlap discrepancies especially in CHM13 cells needs to either be removed without calibration as it is most likely coming from missing calibration. If overlap of peaks should be discussed by authors, please repeat the ChIP with calibration. It is impossible to discuss overlap frequencies without calibration.

Response –

Spike-in controls are especially critical for comparing peak magnitude between samples and conditions to control for variation in IP efficiency, but are less essential for comparing general locations. We appreciate that spike-in controls improve the accuracy in the comparisons of binding of the same proteins across different experiments. While we performed our ChIPs in CHM13 with spike-in controls, the publicly available datasets were all performed without spike-ins. Hence, we have removed comparisons between published cohesin ChIP seq data with our cohesin ChIP seq datasets but have kept comparisons between the locations of cohesin and CENP-A and CTCF. The removal of this information does not change the conclusions in any way.

The 3D SIM is carried out very well. The technique, imaging conditions and analysis are sound and I do not admire the authors for the amount of work it must have been to quantitate all this data. It is important to note that this is by no means routine and that it presents a huge benefit for the presented data.

My main concern with the SIM samples is that they have been mounted in prolong glass, which completely destroys 3D chromatin architecture by flattening the chromosomes entirely (according to our own experiments and rigorous comparison of mounting media for 3D-SIM) This makes this more likely to be 2D-SIM and does not allow accurate 3D measurements. I have to ask the authors to remake the key samples in Slowfade or another suitable non hardening mounting media and reproduce the key observation at least for the supplementary data. I do not ask for extensive quantification, and the spatial exclusion will most likely not change but the measurements need to be in the same ballpark otherwise the exact numbers might be incorrect and mis-lead the research community.

Response –

Inter-CENP-A distances are remarkably comparable when metaphase spreads are imaged using hardening or non-hardening mountants – namely Prolong Glass (562 nm) or Prolong Gold (556 nm) (see Additional figure 1 below). However, we observe substantial loss of resolution when samples are imaged in Prolong Gold. Width (S.D.) of CENP-A clusters go up from 81 nm in Prolong Glass to 108 nm in Prolong Gold. Further, the dip in Rad21 intensity at the center that we detected in our initial experiments disappears when we image the chromosomes in Prolong Gold. Vertical spread (FWHM) of RAD21 between sister chromatids is estimated at 1005 nm (Prolong Gold) in comparison to 792 nm (Prolong Glass). The thickness of RAD21 signal between the sister chromatids is estimated at 168 nm (Prolong Gold) in comparison to 190 nm (Prolong Glass). We believe that the loss in resolution is because the refractive index of Prolong Glass (R.I. – 1.52) is closer to the glass coverslips (R.I. – 1.525) used in this experiment in comparison to Prolong Gold (R.I. – 1.47). Thus, given the trade-off between improved resolution or flattening of chromosomes, we think Prolong Glass medium is a more appropriate choice for this imaging methodology.

Additional Figure 1

Additional Figure 1. Single particle averaging of 3D-SIM images of metaphases mounted in Prolong Gold (non-hardening medium) with aligned CENP-A and RAD21 signals a.

Summed intensities of DAPI, CENP-A and RAD21 around centromeres from $n=303$ chromosomes from 12 mitotic spreads. x and y axes indicate distance (nm). **b-d.** Intensities (white overlay line in A) of CENP-A measured along the lateral axis and RAD21 measured along the lateral and vertical axes are shown (black closed circles in **b-d**), along with their Gaussian fits (magenta line). Dotted lines represent positions of peaks from the fits. The dip in RAD21 intensity at the center of the vertical axis as observed in Figure 3 is not evident upon using Prolong Gold mounting medium.

My second experimental request is that the authors produce controls to prove 3D-SIM resolution in axial and lateral dimension as well as all channels.

Response –

To address the reviewer's request, we performed the following experiment using green beads measuring 100 nm in diameter. The green channel has a shorter wavelength, making it more prone to distortion. Additionally, its resolution is higher, allowing us to detect defects easier. Considering these two aspects, if there is any anomaly in the z dimensions, we should be able to identify it through the green channels.

We arranged these beads in two distinct layers. The first layer was situated on the side of the cover slip, while the second layer was positioned on the side of the microscopy slide. For mounting the slide, we employed Prolong glass. The separation between the two bead layers was 3.41 micrometers. In the subsequent figure (see Additional Figure 2), the term "top" refers to the beads nearer to the cover slip side, and "bottom" denotes the beads closer to the side of the microscopy slide.

To evaluate resolution, both lateral and axial, we utilized the Full Width at Half Maximum (FWHM) method. When assessing lateral resolution, a maximum projection of three slides was performed from both the cover slip side and the microscope slide side. Line profiles with a thickness of three pixels were employed in the calculation, which was subsequently fitted with a single peak Gaussian distribution. The resolution was then quantified using the full width at half maximum. All profiles were normalized to the highest intensity and averaged. A similar approach was applied for the axial resolution. After quantification, there was minimal difference in xy resolution between the two sides. However, the axial resolution on the bottom side displayed a decrease of 12%.

Considering that the measurement range is larger than the size of a chromosome in 3D, we assumed no change in lateral resolution at various steps, with a minor axial resolution change of less than 6%.

Additional figure 2

Additional Figure 2. 3D SIM resolution in lateral and axial dimensions. **a.** Average line profiles in the xy direction for beads placed directly under the surface of the coverslip (top) and at the slide surface (bottom). Separation between the two surfaces is 3.41 μm . **b.** Line profiles were fitted to a single gaussian curve and FWHM from indicated numbers of beads are shown. **c.** Average line profiles in the z direction **d.** Line profiles were fitted to a single gaussian curve and FWHM from indicated numbers of beads are shown.

I disagree that the fact that cohesin is measured to 83 nm makes it likely to be two cohesin complexes. The standard resolution of 3D-SIM is >100 nm and therefore conclusions like these cannot be supported. Furthermore within one channel in 3D-SIM a single cohesin ring with primary and secondary antibody will like be around that size. In addition, conclusions about whether loops are tethered or sisters are encircled should be removed as they are mere speculation given that the molecular mechanisms of either process are still a mystery.

Response –

We agree with the reviewer that conclusions about single or multiple rings of cohesin embracing sister chromatids constitutes overinterpretation and we have removed any mention from the results.

My final concern is that antibodies have been used to obtain all 3D-SIM data. We need to see proof in this study that the antibodies are specific (IF after siRNA or degran) and that they do not exclude each other sterically, which would then produce the observed result. I agree, it is unlikely, but very easy to control, by co-localising RAD21 with SMC1 or similar.

Response –

We tried multiple anti-RAD21 antibodies but only one worked for our purposes. To address the reviewer's concern, we generated an hTERT RPE-1 cell line derivative with the SMC1A gene endogenously tagged with mEGFP using CRISPR-Cas9. SMC1A is one of the three core subunits of human somatic cells and should show identical localization to RAD21. Our intention was to assess endogenous localization of cohesin without signal amplification. We attempted to directly image SMC1A-mEGFP. However, the enrichment of cohesin on mitotic chromosomes is so low that we cannot visualize this without signal amplification using secondary antibodies after the chromosomes are fixed. The enrichment patterns were highly similar for RAD21, visualized by an anti-Rad21 antibody, and SMC1A-GFP visualized with anti-GFP antibody (Fig. 2, 3 and S2a). Both proteins showed enrichment along the sister chromatid axis and formed mutually exclusive enrichment domains with CENP-A. The lateral width (standard deviation) of the RAD21 signal between the two sister chromatids was 190 nm and for SMC1A-mEGFP was 168 nm (Fig. S2b). The main difference was the minor signal external to CENP-A, which was observed for Rad21 but not Smc1A-GFP. As mentioned in the manuscript, this signal was observed by another group using an amplification method. We suspect this binding may be low level and labile.

Finally, the model needs work. It is incredibly messy and unhelpful as the lines are very thick, hard to read annotations half hidden behind drawings.

Response –

We agree and worked with a scientific illustrator to improve the model.

I would not call this approach SPA-SIM. SIM is a microscopy technique and pre-fixes should reflect improvements in the technology. Single particle averaging is a standard procedure used in

imaging and an analysis tool. As described above, it is done excellently but does not need to be hyped by generating yet another acronym.

Response –

The term SPA-SIM has been in the literature for nearly 10 years (Burns et al., 2015). However, it is not important to us to use the acronym so we have adjusted the text throughout the paper to simply describe the methods used.

The paper Woglar et al, 2020 from Anne Villeneuve's lab needs to be cited as this was the first time SIM of a SMC protein on chromosomes was published and it is worth noting and in agreement with the here shown data that Rec8 also localises to the inter-axial space in meiosis.

Response –

This publication is now cited in the introduction.

It would be helpful at the transition from confocal to 3D-SIM to mention that SIM has an 8-fold volumetric resolution increase over confocal microscopy.

Response –

Thank you for this suggestion. We have included this in the text.

In conclusion, I fully support this paper for publication after inclusion of the above-mentioned controls and if they are in line with the currently presented data. The paper is well written, provides a significant advancement in our understanding of mitosis and has potential to be of greater impact to the community by now allowing studies of centromeric cohesion in the 3D chromatin space. Well done!

REVIEWERS' COMMENTS

Reviewer #1 (Remarks to the Author):

The authors have adequately addressed the reviewers comments. I found the revised manuscript much improved.

Reviewer #3 (Remarks to the Author):

Thanks to the authors for addressing the comments.

I understand the RI problem using Prolong Gold. In my opinion, the experiments should have been carried out in non-hardening mounting medium as they concern 3D architecture. It seems this is not easily doable with the current setup of the lab's microscope. I suggest to make clear in methods that hardening Prolong Glass was used as this seems to be the best that can currently been done.

I think its great that the SMC1-EGFP cell line was generated and although imaging of the endogenous locus was not possible in mitosis, the antibody data reproduces RAD21 antibody data. Please make sure in all legends, text and methods when SMC1-EGFP was imaged with GFP antibody. It can easily be explained in methods why this was needed. For future reference, mitotic cohesin images well with Halotag and chemical dyes (much brighter than GFP) and GFP booster can be used for GFP.

All in all, I support this paper for publication as soon as the small textual changes suggested above have been integrated. Good job and nice work for the SIM and cell division fields!

We thank the reviewers for their positive feedback on our findings. We also want to thank reviewer #2 for their useful suggestions on imaging cohesin in our future studies. In response to reviewer #2's remarks, we have incorporated the following revisions in the text –

1. We have included in the Methods section an explanation for the usage of hardening Prolong Glass mounting media for our study.
2. We have included in the Methods section that SMC1-mEGFP was detected using anti-GFP antibody and explained why this was necessary. We have also more clearly depicted in Figure S2a and legends, the use of anti-GFP for the detection of SMC1A-mEGFP.

REVIEWERS' COMMENTS

Reviewer #1 (Remarks to the Author):

The authors have adequately addressed the reviewers comments. I found the revised manuscript much improved.

Reviewer #3 (Remarks to the Author):

Thanks to the authors for addressing the comments.

I understand the RI problem using Prolong Gold. In my opinion, the experiments should have been carried out in non-hardening mounting medium as they concern 3D architecture. It seems this is not easily doable with the current setup of the lab's microscope. I suggest to make clear in methods that hardening Prolong Glass was used as this seems to be the best that can currently be done.

I think its great that the SMC1-EGFP cell line was generated and although imaging of the endogenous locus was not possible in mitosis, the antibody data reproduces RAD21 antibody data. Please make sure in all legends, text and methods when SMC1-EGFP was imaged with GFP antibody. It can easily be explained in methods why this was needed. For future reference, mitotic cohesin images well with Halotag and chemical dyes (much brighter than GFP) and GFP booster can be used for GFP.

All in all, I support this paper for publication as soon as the small textual changes suggested above have been integrated. Good job and nice work for the SIM and cell division fields!